# OFFLINE-TO-ONLINE REINFORCEMENT LEARNING WITH CLASSIFIER-FREE DIFFUSION GENERATION

## ABSTRACT

Offline-to-online Reinforcement Learning (O2O RL) aims to perform online fine-tuning on an offline pre-trained policy to minimize costly online interactions. Existing methods have used offline data or online data to generate new data for data augmentation, which has led to performance improvement during online fine-tuning. However, they have not fully analyzed and utilized both types of data simultaneously. Offline data helps prevent agents from settling too early on suboptimal policies by providing diverse data, while online data improves training stability and speeds up convergence. In this paper, we propose a data augmentation approach, **C**lassifier-**F**ree **D**iffusion **G**eneration (CFDG). Considering the differences between offline data and online data, we use conditional diffusion to generate both types of data for augmentation in the online phase, aiming to improve the quality of sample generation. Experimental results show that CFDG outperforms replaying the two data types or using a standard diffusion model to generate new data. Our method is versatile and can be integrated with existing offline-to-online RL algorithms. By implementing CFDG to popular methods IQL, PEX and APL, we achieve a notable 15% average improvement in empirical performance on the D4RL benchmark like MuJoCo and AntMaze.

## 1 INTRODUCTION

Traditionally, Reinforcement Learning (RL) (Haarnoja et al., 2018) is considered as a paradigm for online learning, where agents learn from online interactions with the environment. Due to costly online interactions in some real-world applications, offline RL (Levine et al., 2020) is proposed where agents learn from a static dataset which is pre-collected by arbitrary policies. Current research in offline RL focuses primarily on addressing the challenge of distribution mismatch or out-of-distribution (OOD) actions through the implementation of a pessimistic update scheme (Kumar et al., 2020) or in combination with imitation learning (Kumar et al., 2019). However, when dealing with a fixed and suboptimal dataset, it becomes exceedingly challenging for offline RL to attain the optimal policy (Kidambi et al., 2020).

Some recent work addresses the above issues employing an offline-to-online setting. Such methods (Lee et al., 2022; Nair et al., 2020) focus on pre-training a policy using the offline dataset and fine-tuning the policy through further online interactions. The main task of O2O RL is to improve sample efficiency and performance. In addition, current methods (Zheng et al., 2023; Nakamoto et al., 2024) have considered the different advantages of offline and online data and utilised both well. In particular, offline data can deter agents from prematurely converging to suboptimal policies due to the diversity of available data, while online data can contribute to training stability and accelerate convergence (Nair et al., 2020; Thrun & Littman, 2000).

However, online data is often limited in the traditional O2O RL setting, which will restrict the exploration of agent in fine-tuning stage. We consider using generative models for data augmentation, rather than simply reusing offline and online data. Prior work has considered upsampling online data with VAEs or GANs (Huang et al., 2017; Imre, 2021). Synthetic Experience Replay (SynthER) (Lu et al., 2024) employs diffusion for data generation, augmenting offline data in offline RL and online data in online RL. When applied directly to O2O RL settings, SynthER can effectively augment online data during the online phase, but overlooks the crucial role of offline data in O2O RL. To fully leverage offline data, Energy-guided Diffusion Sampling (EDIS) (Liu et al., 2024) generates

data aligned with the online policy based on the offline dataset to address the issue of distribution shift. However, online data better aligns with the online policy, yet EDIS uses the offline dataset as input for the diffusion model, which is counterintuitive. Therefore, studying the relationship between offline and online data and determining how to augment these two types of data are crucial problems that need to be addressed.

To fully leverage offline and online data, we analyzed the distributions of these two types of data and applied data augmentation to each separately. We propose our method, **C**lassifier-**F**ree **D**ffusion **G**eneration (CFDG). Our conditional diffusion model requires only a single training session to simultaneously sample new offline and online data, greatly reducing time costs and improving sampling efficiency. To summarize, the contributions of this paper are:

- We analyzed the distributions of offline and online data used in O2O RL and also studied the data distribution generated by the existing method EDIS. We found that performing data augmentation separately for online and offline data yields better results.
- We use conditional diffusion with classifier-free guidance, taking online data and offline data as inputs to generate new samples of both types. This approach avoids overlap between the distributions of the two data types and enhances sample quality without incurring additional training time costs.
- We conducted experiments on the Locomotion and AntMaze tasks, demonstrating that our data augmentation method significantly improves multiple O2O RL algorithms, including IQL, PEX, and APL. Furthermore, our approach outperforms existing data augmentation methods such as SynthER and EDIS.

## 2 PRELIMINARIES

We represent the environment as a Markov decision process (MDP) defined by a tuple $(\mathcal{S}, \mathcal{A}, P, R, \rho_0, \gamma)$, where $\mathcal{S}$ is the state space, $\mathcal{A}$ is the action space, $P(s' \mid s, a)$ is the transition distribution, $\rho_0$ is the initial state distribution, $R(s, a)$ is the reward function and $\gamma \in (0, 1)$ is the discount factor. The objective of RL agent is to find a policy $\pi(a \mid s)$ that maximizes the expected cumulative return $\mathbb{E}_\pi[\sum_{t=0}^{\infty} \gamma^t r_{t+1}]$

### 2.1 OFFLINE REINFORCEMENT LEARNING

In Offline RL, the agent can only access a static dataset $\mathcal{D}$ collected by a behavior policy $\pi_\beta(a \mid s)$. Off-policy RL approaches can take advantage of the offline dataset to train a critic network (Q-function) $Q_\theta^\pi(s, a)$ with parameters $\theta$, which estimates the long-term discounted reward achieved by executing action $a$ in state $s$ and following the policy $\pi$ thereafter. The critic network can be trained using the following temporal difference(TD) learning objective:

$$\mathcal{L}_Q(\theta) = \mathbb{E}_{(s,a,r,s')\sim\mathcal{D}}\left[\left(r + \gamma Q_{\hat{\theta}}\left(s', \pi_\phi\left(s'\right)\right) - Q_\theta(s, a)\right)^2\right], \tag{1}$$

where $\hat{\theta}$ denotes the target value network for stabilizing the learning process. Then, the policy can be updated to maximize the current Q value:

$$\mathcal{L}_\pi(\phi) = \mathbb{E}_{s\sim\mathcal{D}}\left[-Q_\theta(s, \pi_\phi(s))\right], \tag{2}$$

Since $\pi_\phi(s')$ is potentially out of the distribution, $Q_\theta$ could give an incorrect value, resulting in suboptimal policies. To mitigate the well-known extrapolation error in value networks for OOD actions (Fujimoto et al., 2019; Kumar et al., 2020), offline RL methods typically constrain the policy to perform actions close to the dataset through policy constraint (Fujimoto et al., 2019; Fujimoto & Gu, 2021; Kumar et al., 2019), value regularization (Kumar et al., 2020; An et al., 2021), etc. One representative offline RL method is TD3+BC (Fujimoto & Gu, 2021). TD3+BC adds a behavior cloning (BC) regularization term to the standard policy improvement in TD3:

$$\mathcal{L}_\pi^{\text{TD3+BC}}(\phi) = \mathcal{L}_\pi(\phi) + \lambda_{\text{BC}}\mathbb{E}_{(s,a)\sim\mathcal{D}}\left[\left(\pi_\phi\left(s\right) - a\right)^2\right], \tag{3}$$

where $\lambda_{\mathrm{BC}}$ balances the standard policy improvement loss and BC regularization. We can summarize offline RL methods by introducing an additional regularizer to the online RL objective (Guo et al., 2023).

## 2.2 OFFLINE-TO-ONLINE REINFORCEMENT LEARNING

Building upon the concepts of offline RL, offline-to-online RL aims aims at enhancing performance by fine-tuning pre-trained offline policy, which contains two phases: (i) *offline pre-training*, where offline datasets are used to pre-train the policy, and (ii) *online fine-tuning*, where online interactions are used to refine the pre-trained policy.

Currently, in O2O RL algorithms, there are two paradigms for utilizing offline data and online data. (i) An approach is to simply set the ratio of offline data to offline data to 1:1, with each batch containing half of online data and half of offline data, such as PEX (Zhang et al., 2023) and Cal-QL (Nakamoto et al., 2024). (ii) Another approach utilizes an online-offline replay buffer (OORB) in APL (Zheng et al., 2023) and SUNG (Guo et al., 2023), with each batch having a probability $p$ of containing online data and a probability $1 - p$ of containing offline data. As mentioned in Equation (3), $\lambda$ is a trade-off coefficient and decides whether we use the regularizer. When data is sampled from the online buffer, $\lambda$ is set to 0, otherwise 1. Formally, this strategy can be explained as below:

$$\lambda \leftarrow \begin{cases} 0 & \text{if } (\mathbf{s}, \mathbf{a}) \sim \text{online buffer} \\ 1 & \text{otherwise.} \end{cases} \tag{4}$$

## 2.3 DIFFUSION MODELS

Diffusion models create new data by starting with random noise and gradually turning it into something meaningful, like an image or text. Diffusion models (Ho et al., 2020) assume $p_\theta(x_0) := \int p_\theta(x_{0:T}) dx_{1:T}$, where $x_1, \ldots, x_T$ are latent variables of the same dimensionality as the data $x_0 \sim p(x_0)$. A forward diffusion chain gradually adds noise to the data $x_0 \sim q(x_0)$ in $T$ steps with a pre-defined variance schedule $\beta_i$, expressed as

$$q(x_{1:T} \,|\, x_0) := \prod_{t=1}^{T} q(x_t \,|\, x_{t-1}), \quad q(x_t \,|\, x_{t-1}) := \mathcal{N}(x_t; \sqrt{1 - \beta_t} x_{t-1}, \beta_t I).$$

A reverse diffusion chain, constructed as $p_\theta(x_{0:T}) := \mathcal{N}(x_T; \mathbf{0}, I) \prod_{t=1}^{T} p_\theta(x_{t-1} \,|\, x_t)$, is then optimized by maximizing the evidence lower bound defined as $\mathbb{E}_q[\ln \frac{p_\theta(x_{0:T})}{q(x_{1:T} \,|\, x_0)}]$ (Blei et al., 2017). After training, sampling from the diffusion model consists of sampling $x_T \sim p(x_T)$ and running the reverse diffusion chain to go from $t = T$ to $t = 0$. Diffusion models can be straightforwardly extended to conditional models by conditioning $p_\theta(x_{t-1} \,|\, x_t, c)$.

## 2.4 CLASSIFIER-FREE GUIDANCE

In practical scenarios, there is a growing demand to condition the generation on a label $c$. For example, in image synthesis, diffusion models can generate images consistent with input prompts. To address this requirement, classifier guidance (Dhariwal & Nichol, 2021) incorporates an auxiliary classifier $p_\phi(c|x_t)$ to guide the sampling in each reverse denoising step, thereby increasing the likelihood of $c$ given $x_t$. While this method has demonstrated some performance improvements, training a robust classifier for all reverse steps, particularly for the highly noisy input at the initial step, poses a significant challenge and incurs additional training costs.

To avoid training a separate classifier model, classifier-free guidance (Ho & Salimans, 2022) takes c as another input of the denoising neural network to model the conditional diffusion score, i.e., $\epsilon_\theta(x_t, c, t) \approx -\sigma_t \nabla_{x_t} \log p(x_t|c)$ while the unconditional score $\epsilon_\theta(x_t, t)$ is jointly estimated by randomly dropping the text prompt with a certain probability at each training iteration. Then the gradients for the classifier $p_\phi(c|x_t)$ can be estimated as:

$$\nabla_{x_t} \log p(c|x_t) = \nabla_{x_t} \log p_\theta(x_t|y) - \nabla_{x_t} \log p_\theta(x_t)$$
$$= -\frac{1}{\sigma_t}(\epsilon_\theta(x_t, c, t) - \epsilon_\theta(x_t, t)). \tag{5}$$

Along this line, the corresponding diffusion score can be derived as:

$$\hat{\epsilon}_\theta(x_t.c, t) = \epsilon_\theta(x_t, t) + w(\epsilon_\theta(x_t, c, t) - \epsilon_\theta(x_t, t)), \tag{6}$$

where $w$ is set as a global scalar parameter to control the guidance degree of the condition.

# 3 CLASSIFIER-FREE DIFFUSION GENERATION

In order to perform data augmentation for different types of data, in this section, we first visualize the data distributions of offline and online data in O2O RL and analyze the data generated by EDIS. Given the differences between offline data and online data, we utilize conditional diffusion and classifier-free guidance to generate the required data. We train the diffusion model using both offline and online data as inputs, allowing us to sample data with these two different labels.

## 3.1 DISTRIBUTION ANALYSIS OF OFFLINE DATA AND ONLINE DATA

To effectively perform data augmentation in O2O RL, it is essential to conduct a detailed analysis of the distributions of both offline and online data. We conducted experiments on the generative model EDIS (Liu et al., 2024). We utilized t-SNE (Van der Maaten & Hinton, 2008) to perform a visual analysis of three types of data: the offline dataset, the online data collected during the online fine-tuning process, and the data generated using the offline dataset by .

As depicted in Figure 1, the offline data is more evenly distributed, while the online data is more dispersed. The generated data primarily learns from the offline data and is adjusted towards the current online policy using energy guidance. As a result, the generated data retains most of the characteristics of the offline data while also exhibiting some similarities to the online data. This is why EDIS opts to replace the offline data with the generated data during the agent training process, using the generated data in conjunction with the online data. This approach differs from previous work, which trains with both offline and online data directly. Intuitively, it is the part of generated data that aligns with the online policy that led to the performance improvement. EDIS aims to fully utilize offline data and generate data that aligns better with the current online policy. However, in theory, online data itself is more aligned with the current online policy. In the Section 4.2, we also found that SynthER's direct use of online data for generation yields better results than EDIS.

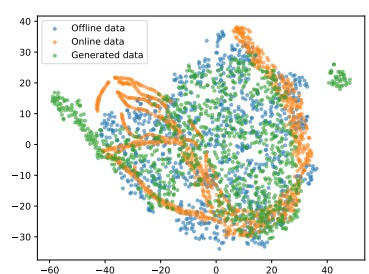

Figure 1: Distribution of offline data, online data and generated data visualized with t-SNE.

Based on the above analysis, directly augmenting the online data would yield better results; However, both offline data and online data are important parts of O2O RL. Offline data prevents the agent from converging to suboptimal policies, while online data enhances training stability and accelerates convergence (Nair et al., 2020; Thrun & Littman, 2000). this also raises a question: how can we effectively utilize the existing offline data? A straightforward idea is to simply perform data augmentation on the offline data. To simultaneously augment offline data and online data, we label the two types of data with distinct labels and use conditional diffusion to generate the data.

## 3.2 CLASSIFIER-FREE GUIDANCE SAMPLING

There are currently two main types of conditional diffusion. One is the classifier guidance (Dhariwal & Nichol, 2021), which involves training a separate classifier to guide the diffusion sampling process. The other is classifier-free guidance (Ho & Salimans, 2022), which does not require an additional classifier but instead trains the model using both conditional and unconditional inputs based on labels. For the first method, the classifier is trained using noise-corrupted data produced by the forward process of conditional diffusion model. Consequently, training an additional classifier can be difficult especially when a significant amount of noise is added to the clean data. Classifier-free guidance is currently the more mainstream approach for diffusion models because it can address the above issues.

We built on the original code implementation of classifier-free guidance, incorporating it into the Elucidated Diffusion Model Karras et al. (2022). We choose to train an unconditional diffusion model $p_\theta(\mathbf{z})$ parameterized through a score estimator $\epsilon_\theta(\mathbf{z}_\lambda)$ together with the conditional model $p_\theta(\mathbf{z}|\mathbf{c})$ parameterized through $\epsilon_\theta(\mathbf{z}_\lambda, \mathbf{c})$. We use a single neural network to parameterize both models, where for the unconditional model we can simply input a null token $\varnothing$ for the class identifier $\mathbf{c}$ when predicting the score, i.e. $\epsilon_\theta(\mathbf{z}_\lambda) = \epsilon_\theta(\mathbf{z}_\lambda, \mathbf{c} = \varnothing)$. We jointly train the unconditional and conditional models simply by randomly setting $\mathbf{c}$ to the unconditional class identifier $\varnothing$ with some probability $p_{\text{uncond}}$, set as a hyperparameter. We then perform sampling using the following linear combination of conditional and unconditional score estimates:

$$\tilde{\epsilon}_\theta(\mathbf{z}_\lambda, \mathbf{c}) = (1 + w)\epsilon_\theta(\mathbf{z}_\lambda, \mathbf{c}) - w\epsilon_\theta(\mathbf{z}_\lambda), \tag{7}$$

where $w$ is a parameter that controls the strength of the classifier guidance. $\tilde{\epsilon}_\theta$ is constructed from score estimates that are non-conservative vector fields due to the use of unconstrained neural networks, so there in general cannot exist a scalar potential such as a classifier log likelihood for which $\tilde{\epsilon}_\theta$ is the classifier-guided score.

To apply it in the O2O RL setting, we use the classifier-free diffusion model to augment both online data and offline data during the online phase. Algorithm 1 describes the classifier-free guidance sampling process in detail. To reduce the high time cost of the construction of diffusion model and data generation, a parameter $T_{\text{diff}}$ is set to perform data augmentation every a certain period of time. First, we will update our diffusion model $M$ using samples from the offline buffer $D_{\text{off}}$ and online buffer $D_{\text{on}}$. Then the diffusion model $M$ will sample synthetic data and add them to synthetic buffer $D_{\text{off\_syn}}$ and $D_{\text{on\_syn}}$. After data generation, we can sample batches from the four buffers mentioned. But how to use different types of data, we need to design different data usage methods for different offline-to-online algorithms.

---

**Algorithm 1** Classifier-free guidance sampling in online phase. Our additions are highlighted in blue.

---

**Input:** Total online fine-tuning steps $T$, pre-trained policy network $\pi_\phi$, pre-trained value network $Q_\theta$, data generation frequency $T_{\text{diff}}$, synthetic data ratio $r \in [0, 1]$
**Initialize:** online buffer $D_{\text{on}} = \emptyset$, offline buffer $D_{\text{off}} \leftarrow$ offline dataset, offline synthetic buffer $D_{\text{off\_syn}} = \emptyset$, online synthetic buffer $D_{\text{on\_syn}} = \emptyset$, conditional diffusion model $M$
**for** $t = 0$ **to** $T$ **do**
    Collect data with $\pi$ in the environment and add to $D_{\text{on}}$
    **if** $t \bmod T_{\text{diff}} == 0$ **then**
        Update conditional diffusion model $M$ with samples from $D_{\text{off}}$ and $D_{\text{on}}$
        Generate offline samples from $M$ and add them to $D_{\text{off\_syn}}$
        Generate online samples from $M$ and add them to $D_{\text{on\_syn}}$
    **end if**
    Train $\pi$ on samples from $D_{\text{on}} \cup D_{\text{off}} \cup D_{\text{off\_syn}} \cup D_{\text{on\_syn}}$ mixed with ratio $r$
**end for**

---

**Data Usage**   O2O RL algorithms usually combine the offline data and online data with a ratio during the training process. Since the addition of synthetic data, another hyperparameter synthetic data ratio $r$ is also required in our framework. In our data augmentation method, synthetic data are sampled from $D_{\text{off\_syn}}$ and $D_{\text{on\_syn}}$ and we can treat the generated offline data and online data as a whole.

According to the two data usage diagrams mentioned in Section 2.2, we design two new data usage methods for synthetic data. For the first paradigm which just combine 50% online data and 50% offline data, we use a simple method which concatenates synthetic data with sampled online data and offline data. The synthetic data ratio $r$ represents the proportion of synthetic data in each batch. Each batch will be concatenation of online data, offline data and synthetic data.

For the second diagram for utilizing data, online data and offline data are sampled from OORB following a Bernoulli distribution. With a probability $p$, data are sampled from the online buffer, and with probability $1 - p$, they are sampled from the offline buffer. Besides, when data is sampled from the online buffer, $\lambda$ is set to 0, otherwise 1, as the Equation 4 shows. Since we add synthetic data into

framework, we use a simple method where synthetic data will be seen as part of online data or offline data and be used for training as the online data or offline data do. The synthetic data ratio $r$ also used to represent the proportion of synthetic data in each batch. Each batch will be concatenation of online data and synthetic data or offline data and synthetic data.

## 4 EXPERIMENTS

In this section, we show the efficiency of our CFDG method through empirical validation. Section 4.1 commence by showcasing its excellent performance on the D4RL benchmark (Fu et al., 2020) and also shows generalizability and statistical significance on baselines like IQL (Kostrikov et al., 2021), PEX (Zhang et al., 2023) and APL (Zheng et al., 2023). Section 4.2 compares our method, CFDG, with other model-based approaches such as SynthER (Lu et al., 2024) and EDIS (Liu et al., 2024), highlighting the superiority of our conditional diffusion model over other diffusion models. In Section 4.3, we performed an ablation study to examine two key components of our method, generating online data and generating offline data using CFDG. Both were shown to effectively improve the performance of the algorithm.

### 4.1 OFFLINE-TO-ONLINE RL EXPERIMENTS

**Datasets** Our method is validated on two D4RL (Fu et al., 2020) benchmarks: Locomotion and AntMaze, which are used by IQL (Kostrikov et al., 2021) and PEX (Zhang et al., 2023). Locomotion includes diverse environment datasets collected by varying quality policies. We assess algorithms on hopper, halfcheetah, and walker2d environment datasets, each with four quality levels. AntMaze tasks involve guiding an ant-like robot in mazes of three sizes (umaze, medium, large), each with two different goal location datasets. We focus on the two larger size mazes (medium, large) which have the lowest offline performance (as opposed to, e.g., using the umaze, which leaves little room for further improvement). The evaluation environments are listed in Table 1's first column.

**Baselines** We consider the following baselines (i) **IQL** (Kostrikov et al., 2021) learns a value network to match the expectile of the critic network to address out of distribution problem. (ii) **PEX** (Zhang et al., 2023) freezes the pre-training policy and introduces policy expansion to enhance exploration. (iii) **APL** (Zheng et al., 2023) leverages the distinct advantages of offline and online data for adaptive constraints. These three are relatively new O2O RL algorithms and cover two paradigms of data utilization. According to Section 2.2, IQL and PEX follow the first paradigm, while APL follows the second, allowing us to test the generality of our algorithm.

We use the original paper's implementation for all three baselines. Every experiment starts with training a model only using the offline dataset, the same as in the original work. We note that, since APL did not conduct experiments on the AntMaze dataset, we carried out our experiments according to its original setup.

**Settings** For IQL and PEX, we perform 1M update steps for offline pre-training and then 1M environment steps for online fine-tuning. For APL, we perform 1M pre-training steps and 0.1M fine-tuning steps to ensure consistency with the original paper. For our data augmentation method, the synthetic buffer size is set to 1M. The data generation frequency $T_{\text{diff}}$ as 10K in APL and 100K in IQL and PEX. The generated data ratio $r$ is set to $1/3$. Therefore, the percentage of online data, offline data and generated data will be $1 : 1 : 1$. In the generated data, the ratio of generated online data to generated offline data is $8 : 2$. The above configurations keep the same across all tasks, datasets and methods.

As shown in Table 1, the integration of CFDG outperforms all baselines. Using diffusion model with classifier-free guidance to generate offline data and online data can surpass the baseline algorithm on over 10 datasets in locomotion tasks and 4 datasets in antmaze tasks. IQL and PEX achieve a 15% average improvement in empirical performance and APL achieve a 11% average improvement.

### 4.2 COMPARISONS BETWEEN CFDG AND MODEL-BASED METHODS

In addition to demonstrating the effectiveness of incorporating CFDG into standard O2O RL algorithms, we also aim to prove that CFDG outperforms current SOTA data augmentation methods. We

Table 1: **Enhanced performance achieved by CFDG after online fine-tuning on Locomotion and Antmaze tasks.** We evaluate the normalized score of standard base algorithms (including IQL (Kostrikov et al., 2021), PEX (Zhang et al., 2023) and APL (Zheng et al., 2023), denoted as "Base") in comparison to the base algorithms augmented with CFDG (referred to as "Ours"). All results are assessed across 5 random seeds. The superior scores are highlighted in blue .

| Dataset[1] | IQL | | PEX | | APL | |
|---|---|---|---|---|---|---|
| | Base | Ours | Base | Ours | Base | Ours |
| halfcheetah-r-v2 | 53±6 | 65 ± 3 | 78±2 | 81 ± 7 | 93±8 | 103 ± 7 |
| halfcheetah-mr-v2 | 54±0 | 65 ± 2 | 68±3 | 83 ± 3 | 76±40 | 96 ± 2 |
| halfcheetah-m-v2 | 69±2 | 75 ± 1 | 78±5 | 87 ± 3 | 77±39 | 86 ± 28 |
| halfcheetah-me-v2 | 95 ± 1 | 93±1 | 90±3 | 93 ± 0 | 96±3 | 98 ± 2 |
| hopper-r-v2 | 16 ± 13 | 10±1 | 8±0 | 8 ± 0 | 51 ± 30 | 30±40 |
| hopper-mr-v2 | 66±33 | 86 ± 30 | 66±25 | 83 ± 20 | 88±29 | 100 ± 15 |
| hopper-m-v2 | 93±6 | 97 ± 7 | 91±30 | 100 ± 7 | 103 ± 2 | 99±11 |
| hopper-me-v2 | 68±28 | 103 ± 17 | 74±22 | 94 ± 19 | 104±10 | 112 ± 1 |
| walker2d-r-v2 | 15±8 | 18 ± 16 | 18±10 | 65 ± 37 | 12±11 | 27 ± 42 |
| walker2d-mr-v2 | 81±17 | 108 ± 2 | 101±7 | 112 ± 12 | 70±35 | 109 ± 12 |
| walker2d-m-v2 | 88±7 | 96 ± 4 | 101±8 | 108 ± 5 | 92±26 | 102 ± 21 |
| walker2d-me-v2 | 113±0 | 118 ± 3 | 116 ± 1 | 111±4 | 111±1 | 120 ± 10 |
| **locomotion total** | 810 | 933 | 890 | 1024 | 972 | 1081 |
| antmaze-medium-play-v2 | 82±13 | 76 ± 5 | 82±13 | 88 ± 11 | – | – |
| antmaze-medium-diverse-v2 | 82±10 | 86 ± 5 | 90±14 | 98 ± 5 | – | – |
| antmaze-large-play-v2 | 48±13 | 52 ± 18 | 54±19 | 56 ± 21 | – | – |
| antmaze-large-diverse-v2 | 38±16 | 52 ± 22 | 38±16 | 42 ± 16 | – | – |
| **Antmaze total** | 250 | 266 | 264 | 284 | | |

[1] r: random, mr: medium-replay, m: medium, me: medium-expert.

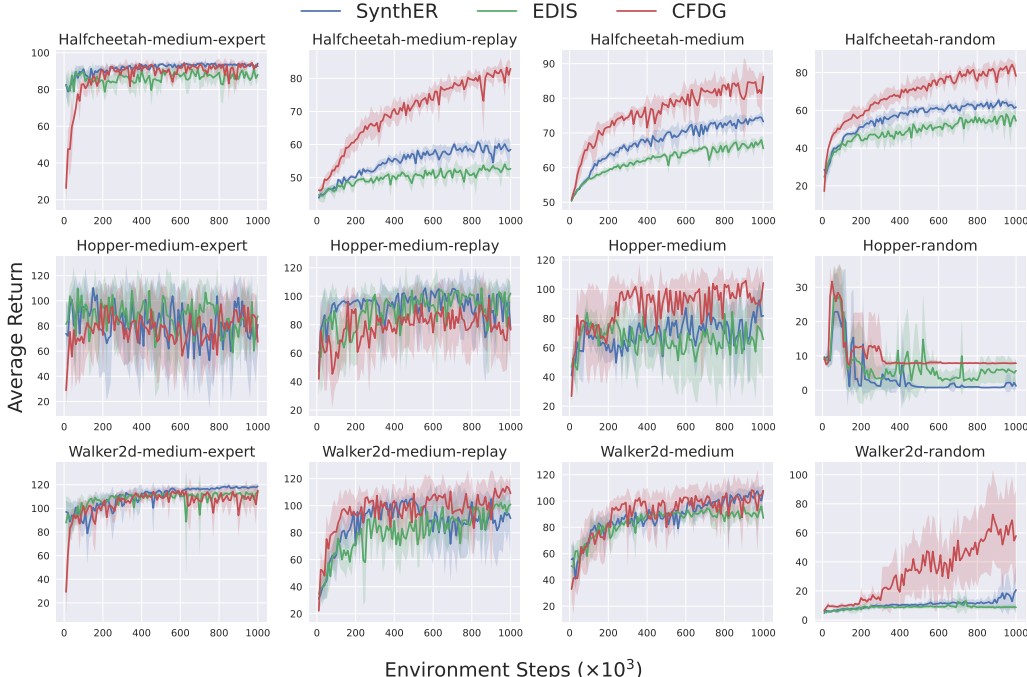

Figure 2: Learning curves of base algorithm augmented with model-based methods SynthER, EDIS and CFDG. Results are averaged over 5 random seeds.

compare CFDG with two model-based methods SynthER (Lu et al., 2024) and EDIS (Liu et al., 2024). In O2O RL, SynthER directly uses a diffusion model to augment online data, while EDIS employs an energy-guided diffusion model to generate new data from offline data. The key difference between

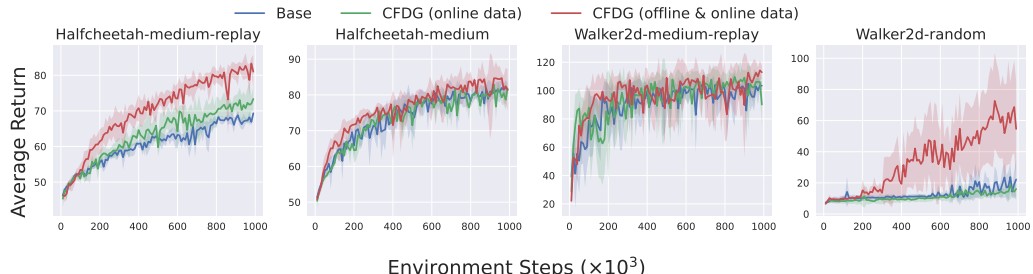

Figure 3: Learning curves of base algorithm augmented with online data and augmented with both offline & online data using our CFDG. Results are averaged over 5 random seeds.

CFDG and these two methods is that CFDG separates offline data and online data into two distinct labels and uses a conditional diffusion model for generation. By accounting for the differences in the distributions of these two types of data and their distinct roles in O2O RL, performing data augmentation for each separately can significantly enhance the algorithm's performance. The base algorithm is IQL and the results are shown in Figure 2.

Based on the experimental results, SynthER performs better than EDIS when directly augmenting online data, which aligns with the intuition that online data is more aligned with the current policy. However, using CFDG to augment both online and offline data simultaneously further improves performance. This is particularly evident in the halfcheetah environment, where CFDG shows significant improvement compared to SynthER and EDIS.

## 4.3 ABLATION STUDIES

The two main differences between the CFDG method and existing model-based approaches are: (i) the diffusion model utilizes classifier-free guidance (ii) it performs data augmentation on both offline data and online data. Therefore, we need to demonstrate that both components effectively enhance performance. We conduct ablation studies on each components in hopper-medium-replay-v2, halfcheetah-medium-v2, walker2d-medium-replay-v2 and walker2d-random-v2.

As shown in Figure 3, using CFDG to augment online data outperforms the baseline. Moreover, augmenting both offline and online data simultaneously leads to further performance improvements, especially on the halfcheetah-medium-replay-v2 and walker2d-random-v2 datasets, where our method significantly boosts the original baseline performance.

## 5 RELATED WORK

**Offline-to-online Reinforcement Learning** Offline-to-online RL aims to improve suboptimal offline policies through online fine-tuning. Prior work usually improve the agent by adding regularizer to mitigate the distribution shift problem or leverage both offline data or online data in online fine-tuning. Kostrikov, Nair, and Levine (Kostrikov et al., 2021) introduce IQL, which incorporates a weighted behavioral cloning step to enhance online policy improvement and is applicable in both online and offline-to-online scenarios. When online interactions are available, such conservative designs may adversely affect the performance. OFF2ON (Lee et al., 2022) employs a balanced replay scheme to address the distribution shift issue. It uses offline data by only selecting near-on-policy samples. However, practical scenarios may involve agents pretrained by various offline RL algorithms, highlighting the necessity for developing a generic offline-to-online RL framework. Recent studies place a growing emphasis on adaptability. PEX (Zhang et al., 2023) freezes the pre-trained policy and initializes a random policy to enhance exploration. PROTO (Li et al., 2023) gradually evolves the regularization term to relax the constraint strength. From a data-centric perspective, APL (Zheng et al., 2023) and SUNG (Guo et al., 2023) impose constraints exclusively on data from offline datasets and data with high uncertainty, respectively. Our method also focuses on data utilization, using data augmentation to expand the available data, thereby enabling more comprehensive learning and improving the agent's performance.

**Diffusion Models in RL** Pearce et al. (Pearce et al., 2023) propose using diffusion model to better imitate human behaviors due to their expressiveness and stability. Diffuser (Janner et al., 2022) applies a diffusion model as a trajectory generator, where the full trajectory of state-action pairs form a single sample for the diffusion model. Additionally, a separate return model is trained to predict the cumulative rewards of each trajectory sample, and its guidance is incorporated into the reverse sampling stage. This approach is similar to Decision Transformer (Chen et al., 2021), which also learns a trajectory generator through GPT2 with the help of the true trajectory returns. However, when used online, sequence models can no longer predict actions from states autoregressively since the states are an outcome of the environment. Consequently, during evaluation, a whole trajectory is predicted for each state but only the first action is applied, leading to high computational cost. Our approach employs diffusion models to do data augmentation for RL in a distinct manner.

## 6 CONCLUSION

In this paper, we analyze the distributions of offline and online data in the O2O RL setting and improve upon existing data augmentation methods by addressing their limitations. We use a diffusion model guided by classifier-free guidance to simultaneously augment both types of data. In the online phase, we input offline data and online data as two distinct labels into the diffusion model. With a single round of training, we can sample both types of data. Our method, CFDG, is simple and can be easily integrated with existing O2O RL algorithms, significantly boosting their performance while surpassing other data augmentation methods.

Although our method achieved the expected performance, there are still some limitations. One of these is the setting of the data ratio. In our experiments, we used a fixed parameter for this, which produced good results. However, according to some of our tests, the ratio of offline to online data can significantly impact performance in different environments. Additionally, determining the optimal ratio for the three types of data, including the generated data, remains an open challenge.

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
