# OpenReview forum: "Offline-to-Online Reinforcement Learning with Classifier-Free Diffusion Generation"
_ICLR.cc/2025/Conference — Submitted to ICLR 2025_

### Official Review · Reviewer_op9D · 2024-10-21

**Soundness:** 2
**Presentation:** 2
**Contribution:** 1
**Rating:** 3
**Confidence:** 4

**Summary:**

To enhance offline-to-online reinforcement learning algorithms, this paper proposes a data augmentation approach called Classifier-Free Diffusion Generation (CFDG). Recognizing the differences between offline and online data, the authors use conditional diffusion to generate both types of data for augmentation during the online phase. This approach aims to improve the quality of sample generation.

**Strengths:**

The method is designed with simplicity in mind, using a motivating example to illustrate the data distribution and introduce the proposed approach.

**Weaknesses:**

1. Incomplete method description of hyperparameters, requiring significant adjustment for implementation.
2. Incomplete sentence, like on line 181.
3. Limited performance improvements, with potentially misleading labels (e.g., antmaze-medium-play IQL).
4. Lack of novelty, as careful adjustments to existing baselines could yield similar results.

**Questions:**

How can we embed the class identifier when constructing a conditional diffusion model?

---

### Official Review · Reviewer_6gFc · 2024-10-28

**Soundness:** 2
**Presentation:** 2
**Contribution:** 1
**Rating:** 3
**Confidence:** 4

**Summary:**

This paper proposes to utilize classifier-free diffusion for data augmentation. More specifically, it distinguishes the difference between online data and offline data and uses conditional diffusion to generate both types.

**Strengths:**

The motivation is clear

**Weaknesses:**

1. The novelty is limited.
2. The performance improvement is negligible and sensitive to hyperparameters.
3. Ablation studies fail to provide enough insights into the algorithm.

**Questions:**

No

---

### Official Review · Reviewer_VF2E · 2024-11-01

**Soundness:** 2
**Presentation:** 2
**Contribution:** 1
**Rating:** 3
**Confidence:** 4

**Summary:**

This paper introduces CFDG, a method that augments both offline and online data separately to enhance offline-to-online reinforcement learning. Experimental results on benchmarks like D4RL demonstrate that integrating CFDG into standard O2O RL algorithms, such as IQL, PEX, and APL, yields an average 15% improvement over prior data augmentation approaches like SynthER and EDIS. CFDG thus provides an effective and adaptable way to boost O2O RL performance through refined data augmentation.

**Strengths:**

1. By analyzing the distinct distributions of offline and online data, the authors identify the benefits of separately augmenting each data type.
2. The use of conditional diffusion with classifier-free guidance allows the generation of high-quality offline and online samples independently.
3. Through comprehensive experiments on challenging benchmarks like Locomotion and AntMaze, the paper shows that its approach significantly boosts the performance of multiple O2O RL algorithms.

**Weaknesses:**

1.The paper mentions that the ratio of offline to online data and the ratio of real data and synthetic data can significantly impact performance but does not explore this aspect in detail. Specifically, a sensitivity analysis of these ratios would help determine optimal values or provide insights into the adaptability of CFDG in diverse O2O RL scenarios.
2. The paper lacks sufficient innovation; the difference in distribution between offline and online data is obvious, and the analysis in Section 3.1 does not provide any new insights. The classifier-free guided diffusion model used for data generation is also based on existing work. The only innovative aspect of the paper is the separate generation of offline and online data.

**Questions:**

1. The current explanation assumes that separate generation is beneficial due to distribution differences, but this could be expanded by quantifying how these differences affect policy optimization. Could the authors provide additional theoretical or empirical justification for why separate generation of offline and online data leads to significant performance gains in O2O RL?
2. How sensitive is the performance of CFDG to changes in the offline-to-online and synthetic data ratios?

---

### Meta-Review · Area_Chair_7pyE · 2024-12-21

**Metareview:**

This paper presents a data augmentation method for offline-to-online reinforcement learning that separately augments offline and online data. While the paper demonstrates a clear motivation and presents experimental results showing modest improvements over existing methods, all three reviewers identified significant concerns about the limited novelty, incomplete method descriptions, and sensitivity to hyperparameters. The main contribution of separately generating offline and online data, while logical, fails to provide sufficient justification for the observed performance gains.

**Additional Comments On Reviewer Discussion:**

There were no rebuttals, and the reviews are mostly consistent.

---

### Decision · Program_Chairs · 2025-01-22

Reject